# Geometric control of diffusing elements on InAs semiconductor surfaces via metal contacts

Sandra Benter [1,2] ✉, Adam Jönsson[2,3], Jonas Johansson[1,2], Lin Zhu[4], Evangelos Golias [4], Lars-Erik Wernersson [2,3] & Anders Mikkelsen [1,2]

Local geometric control of basic synthesis parameters, such as elemental composition, is important for bottom-up synthesis and top-down device definition on-chip but remains a significant challenge. Here, we propose to use lithographically defined metal stacks for regulating the surface concentrations of freely diffusing synthesis elements on compound semiconductors. This is demonstrated by geometric control of Indium droplet formation on Indium Arsenide surfaces, an important consequence of incongruent evaporation. Lithographic defined Aluminium/Palladium metal patterns induce well-defined droplet-free zones during annealing up to 600 °C, while the metal patterns retain their lateral geometry. Compositional and structural analysis is performed, as well as theoretical modelling. The Pd acts as a sink for free In atoms, lowering their surface concentration locally and inhibiting droplet formation. Al acts as a diffusion barrier altering Pd's efficiency. The behaviour depends only on a few basic assumptions and should be applicable to lithography-epitaxial manufacturing processes of compound semiconductors in general.

Compound semiconductors are central for optoelectronic, mobile, and power technologies and strong candidates for future quantum technologies[1–9]. Synthesis and epitaxy rely on controlling the concentration ratio of each involved element at the growth surface, as well as other parameters like substrate temperature and crystal structure. Local geometric control of the growth is important as it allows bottom-up synthesis in specific regions when manufacturing a complete device on a chip. Various concepts have been developed attempting to resolve this, such as template-assisted growth, metal seed particle-induced growth or the use of surface crystal facet[10–16]. While proven highly useful, these concepts all rely on altering the growth substrate by either blocking or promoting synthesis in specific areas. An interesting alternative is to locally vary the concentration of free (diffusing) synthesis elements on the surface without altering the growth substrate directly, as we pursue in the present work. Using lithographically defined metal stacks for this purpose would be highly desirable as they are already an intimate part of device fabrication in the form of electrical contacts.

Droplet formation and dynamics have fascinated researchers for decades due to the complex behaviour found for even simple systems[17–23]. For a wide range of compounds, droplet formation is observed, including metal alloys, 2D transition metal dichalcogenides, SiC and III-V semiconductors[17–21,24–26]. The creation of metallic droplets of group III elements is widely occurring on III-V compounds when heating the sample above the congruent evaporation temperature. Here, group V elements evaporate more effectively, resulting in an excess of group III atoms on the surface which can agglomerate. Studies have focused on the formation[27,28], appearance[29] and motion[30–34] of Ga and In-droplets on GaAs, GaP or InAs after removing the native oxide. Droplets can be desirable for the growth of quantum dots and nanowires[16,35–43], however.

[1]Department of Physics, Lund University, Box 118, Lund 22100, Sweden. [2]NanoLund Center for Nanoscience, Lund University, Box 118, Lund 22100, Sweden. [3]Department of Electrical and Information Technology, LTH, Box 118, Lund 22100, Sweden. [4]MAX IV Laboratory, Lund University, Box 118, Lund 22100, Sweden. ✉e-mail: sandra.benter@sljus.lu.se

The formation of droplets during device manufacturing steps can also be unwanted as it can lead to poor control of the growth quality and lack of lateral control[15,44,45].

In this paper, we explore the geometric control of the surface concentration of freely diffusing In atoms on the InAs(111)B surface. This is evidenced by a dramatic change of In-droplet formation during annealing, observed around lithographically defined Al/Pd-stacks. This results in droplet-free zones at much higher temperatures than normally possible in vacuum. The behaviour can be controlled by the Al/Pd stack composition and position, as well as annealing time and temperature. The deposited metal stacks remain localised within their original boundaries during all process steps. A theoretical model is developed, which explains our observations based on simple and general assumptions. The As evaporates rapidly at elevated temperatures leaving free In to diffuse on the surface. The Pd-layer acts as a sink for these free In atoms. As a result, the overall In-density around the metal patterns is lowered, impeding the formation of droplets. Figure 1a depicts the concept. Adding Al between the InAs substrate and the Pd reduces the vertical alloying of Pd into the substrate and alters the efficiency of Pd to act as a sink. Finally, we determine limits for this effect as governed by the finite amount of In incorporation possible into the Pd.

The general concept presented here opens up significant opportunities to design manufacturing processes for compound semiconductor devices, pushing up annealing temperatures for non-congruent systems and simultaneously controlling the formation of structures across the surface.

## Results

### Indium droplet formation on InAs near metal patterns as a function of temperature

To observe the influence of metal patterns on the formation of In-droplets on InAs(111)B substrates due to annealing, we implemented circular and square-shaped metal stacks on the surface in varying sizes (diameters and side lengths 20–160 µm). The metal stacks consisted of 5 nm Al topped with 20 nm Pd. The patterns were repeated over the sample (Fig. 1b). Every annealing step was performed in an ultra-high vacuum (UHV) chamber with a base pressure in the $10^{-10}$ to low $10^{-9}$ mbar regime.

For all samples, the native oxide on the surfaces was removed by a standard cleaning method of annealing at 400 °C using atomic Hydrogen for ~20 min[46], after which the Hydrogen was turned off and the temperature lowered to 350 °C. At this point, no droplets were found.

To observe the temperature dependency for In-droplet formation, annealing was carried out on separate samples at 500, 550, 600 and 650 °C with a heating rate of 3 °C/s. The temperature was directly quenched upon reaching the target value. Subsequently, the samples

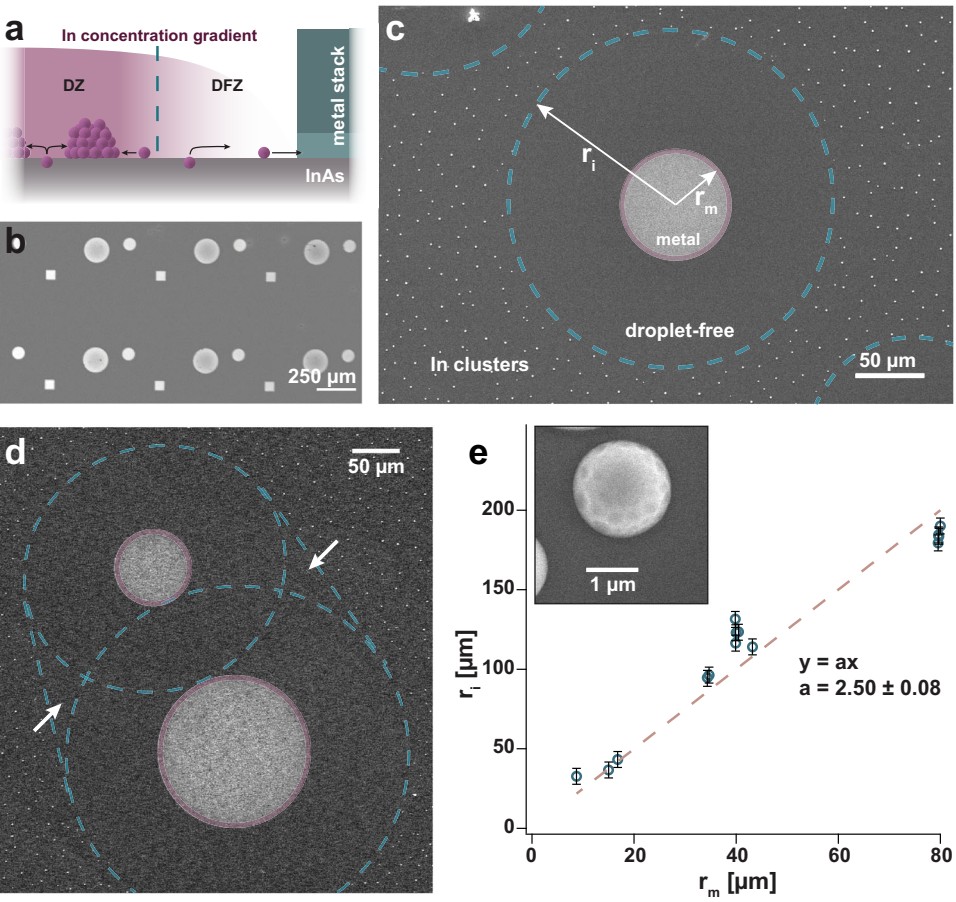

**Fig. 1 | Establishing the droplet-free zone. a** Sketch of the proposed mechanism: In atoms will diffuse on the surface, either forming droplets in the droplet zone (DZ) or diffusing into the metal pattern in the droplet-free zone (DFZ), dependent on their position on the substrate. **b** SEM image of the lithographic pattern after metal evaporation (5 nm Al and 20 nm Pd) and lift-off, **c** SEM image after heating the sample rapidly to a temperature of 600 °C. The coloured lines indicate the three distinct zones forming due to the presence of the metal pattern. **d** Two adjacent metal patterns with coalescent droplet-free zones. The gain in the droplet-free area is indicated by the arrows. **e** Dependency between the radius of individual metal patterns ($r_m$) and the resulting radius of the droplet-free and metal zone ($r_i$) for different-sized square and circular metal stacks. The error bars indicate the estimated, conservative error for determining the size of each DFZ. The inset shows a more detailed image of an In-droplet found after annealing the sample to 600 °C. Source data are provided as a Source Data file.

were studied using Scanning Electron Microscopy (SEM) (see "Methods" for more details). We find that nm-sized droplets start to appear after annealing to 550 °C. µm-sized droplets form at 600 °C and 650 °C. Three distinct areas are found on the substrate: (1) the metal zone (MZ) with the deposited Al/Pd-stack, (2) the droplet-free zone (DFZ) on the InAs(111)B substrate around the metal and (3) the droplet zone (DZ) displaying µm-sized In-droplets further away from the metal stacks. Figure 1c shows an overview of the location of all areas on the sample surface. In the DZ, we observe µm-sized In-droplets forming with an average size of 1.43 ± 0.53 µm after annealing to 600 °C. The size of the DFZ is determined by fitting an ellipse to the border of the surrounding area displaying In-droplets. By calculating the dependency between MZ and DFZ (excluding regions exhibiting droplets), we find that $r_i = 2.5 \cdot r_m$ ($r_i$—radius of the DFZ, $r_m$—of the metal pattern), see Fig. 1e. In general, around 14.0% of the combined DFZ and MZ indicated in Fig. 1c (blue, dashed outline) is covered by the Al/Pd-stack and about 86.0 ± 3.8% belongs to the DFZ. This relation is independent of the shape and size of the metal pattern. For both—circular and square—metal stacks, the DFZ appears to be circular and centred around them; see Supplementary Note 2 for more details. The compact metal stacks employed in our study allow for a direct link between the radii of the individual zones. This provides an easy connection to the theoretical model proposed below. For more complicated or elongated structures, see Supplementary Fig. 3 and the related discussion.

DFZs coalesce and increase slightly (see the blue outlined area in Fig. 1d) when the respective metal stacks are positioned close enough. We do not see any long-range dependencies between different metal stacks when the distance increases further. However, implementing a smart layout of metal structures over a larger area yields the possibility of avoiding any droplet formation for processes involving sample temperatures significantly higher than normally possible.

Finally, when annealing samples to above 650 °C, the DFZ around the metal pattern vanishes (see Supplementary Fig. 1). Instead, droplets, several tens of µm large form, move along the surface and coalesce. As will be discussed below, the disappearance is likely due to an In-saturation of the Pd, eliminating the sink effect.

This explanation would also indicate that prolonged annealing at low temperatures should reduce the ability of the Pd to inhibit droplet formation, which is indeed observed in the Supplementary Video and discussed in Supplementary Note 7.

## Morphology and composition at the metal-semiconductor boundary

Atomic Force Microscopy (AFM) measurements were performed on the border of the Al/Pd-stack and InAs surface before and after annealing to 600 °C (see Fig. 2a–c). It is observed that upon heating, the boundary between the Al/Pd-stack and the InAs stays intact. The size and outline of individual metal patterns appear laterally unaltered on the surface. However, the annealing induces a substantial

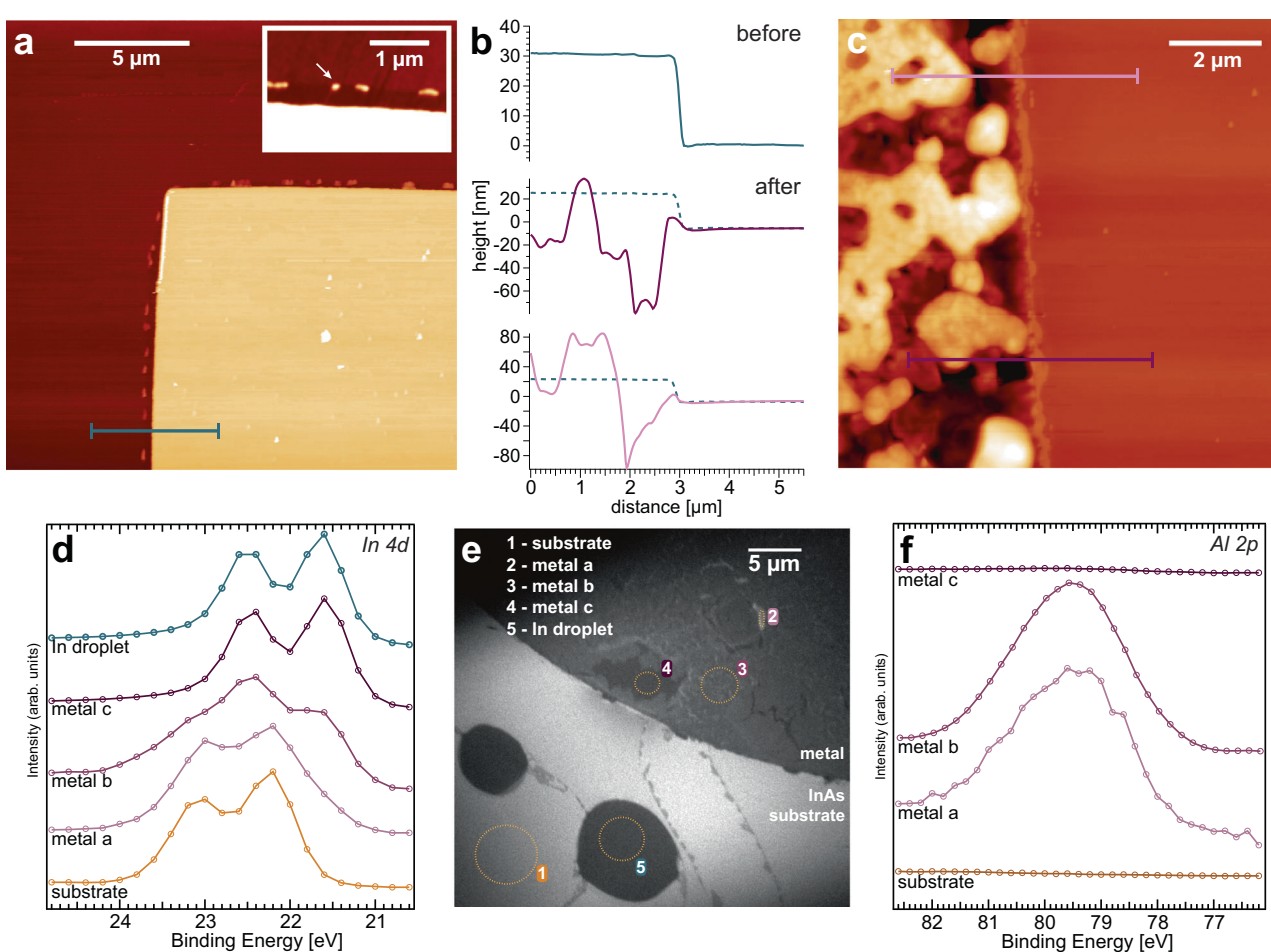

**Fig. 2 | Induced changes at the metal pattern edge.** AFM images at the edge of the metal patch before (**a**) and after removing the native oxide of the substrate and annealing to 600 °C (**c**). Inset in (**a**): close-up of metal edge showing Al protruding below the Pd pattern (white arrow). **b** Shows line scans drawn in (**a**) and (**c**) to compare the height profile before and after annealing. **d** *In 4d* and **f** *Al 2p* core level extracted from measurement positions indicated in (**e**) via XPEEM image analysis. Different domains within the metal stack are visible, indicating various chemical environments. Al is confined to the original position of the lithography pattern. The intensity for all spectra was scaled for better comparison (for the original intensities see Supplementary Fig. 6). Source data are provided as a Source Data file.

morphological transformation inside the metal stack itself. Prior to annealing, its surface is flat with a roughness below a few nm and a total height of 30 nm. After annealing to 600 °C, µm-sized protrusions and holes are formed (up to 80 nm in height variation). Their level is both above and below that of the InAs substrate, respectively, indicating that vertical alloying has occurred with a significant rearrangement in the Al/Pd-stacks.

To gain a better understanding of the chemical and elemental changes induced in the metal and semiconductor, synchrotron-based X-ray Photoemission Electron Microscopy (XPEEM) and µ-X-ray Photoemission Spectroscopy (XPS) measurements were performed in UHV directly after sample treatment. The experimental setup is explained in detail in the "Methods".

XPEEM measurements show the presence of In and As in the metal after annealing, with a substantial variation in both amount and chemical environment (see Supplementary Note 3). Pure In was detected within some areas (core level peaks similar to the droplets found on the InAs substrate), as well as in areas where Pd, Al and As were present (Fig. 2d). While Pd-In alloying is consistent with the binary phase diagram, intermixing of Al and In is not[47,48]. However, both Pd and Al are observable on the surface of the metal, indicating that intermixing of the original Pd- and Al-layer must have occurred (Fig. 2f). Otherwise, only the top Pd-layer would be measurable in surface-sensitive XPS. Thus, our measurements show a large intake of In into the metal as well as intermixing of Al and Pd, which is consistent with the roughening observed in AFM and SEM. In is likely binding primarily to Pd as well as forming areas of pure, metallic In. Although some of it originates from the InAs below the Pd, a significant In-intake from the surrounding surface is possible. We can roughly estimate the intake of In into the Pd of a metal stack (see Supplementary Note 4 for the dominating domain in the field of view), giving a ratio of about 0.1 In to Pd atoms. Importantly, we observe no traces of Al or Pd atoms outside the original metal pattern via µ-XPS after annealing, confirming the lateral integrity of the layout once more.

## Theoretical modelling of the observed DFZ

The formation and shape of the DFZ can be understood based on a few basic assumptions: First, an excess of free In atoms is present on the surface when annealing the sample above the congruent evaporation temperature. Second, alloying of In with Pd is favourable; thus, areas with deposited Pd will act as a sink for free In atoms until saturation occurs. The formation and size of the DFZ can now be modelled by a mass-balance and nucleation approach. We use mass balance to estimate the surface decomposition rate $F$ at which In (and As) atoms are released from the InAs surface during annealing. Due to the low vapour pressure of In[49,50], we assume that the released In stays on the surface and, if close enough to the Al/Pd-stack, diffuses into the Pd. The atomic fraction of In in Pd is given by $x_{In} = N_{In}/(N_{In} + N_{Pd})$, where $N_{In}$ is the number of In and $N_{Pd}$ the number of Pd atoms. The latter is given by $N_{Pd} = \pi r_m^2 h_{pd}\rho_{Pd}$, where $r_m$ is the radius, $h_{Pd}$ the thickness and $\rho_{Pd}$ the number density of the Pd-layer. Similarly, $N_{In} = \pi(r_i^2 - r_m^2)Ft$, where $r_i$ is the radius of the area of interest (depicted in Fig. 1c) and $t$ the time it takes to reach $x_{In}$. Combining both expressions with $x_{In}$ results in the proportionality

$$r_i = r_m\sqrt{\frac{x_{In}}{1-x_{In}}\frac{h_{Pd}\rho_{Pd}}{Ft}+1} \qquad (1)$$

which is experimentally observed and illustrated in Fig. 1e. From the fraction of In in the In/Pd alloy ($x_{In} = 0.1$), the time ($t = 20$ min) and the initial thickness of the Pd-layer ($h_{Pd} = 20\,nm$), we can estimate the surface decomposition rate $F = 2.4 \times 10^{12}\,cm^{-2}s^{-1}$ (see Supplementary Information for experimental values).

To explain and understand the formation of the DFZ, we apply a nucleation approach. A reduced set of rate equations is used[51], where

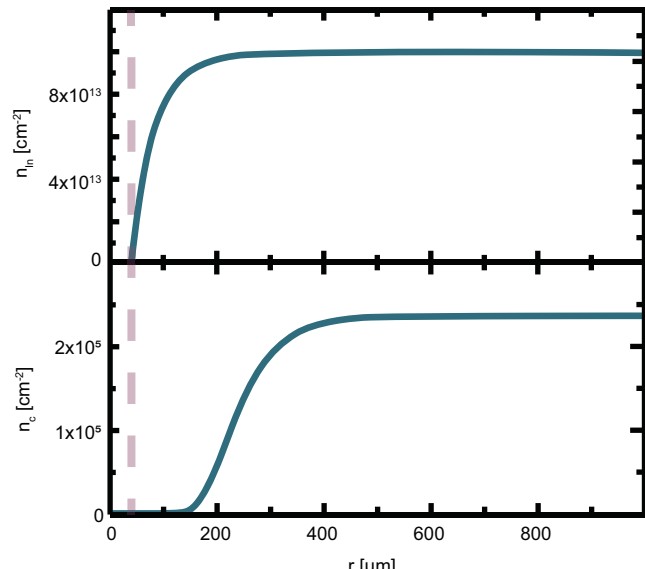

**Fig. 3 | Model for the DFZ.** The surface density of single In atoms $n_{In}$ (upper panel) and In-droplets $n_c$ (lower panel) outside a circular Pd-layer with a radius of 40 µm (marked with the dashed line).

we consider the concentrations of In atoms $n_{In}$ and stable droplets $n_c$. Meanwhile, a continuous flux of In atoms into the Pd is present. As a result, a concentration gradient arises, depleting the area around the metal of In atoms and therefore suppressing the formation of droplets. Subsequently, we observe the formation of the DFZ. The reduced rate equation model is explicitly outlined in Supplementary Note 5. In Fig. 3, we show a fit of the model to the observed $r_i$ and droplet density far away from the Pd-layer of radius $r_m = 40$ µm. It is evident that the droplet density is essentially zero for $r < r_i$ and approaches a constant value for larger $r$ in accordance with our experimental observations.

## Adding functionality by changing the metal stack

The functionality of the lithography pattern can be tuned by modifying the thickness of the layers in the metal stack. To exemplify this, patterns with different Al thickness were implemented. First, we investigate samples with a 15 nm-thick Al-layer topped by 20 nm Pd. The increased thickness of Al suppresses the formation of the DFZ around the metal when annealing to 600 °C. In contrast, it causes an accumulation of larger In-droplets on the Al rim along the Pd pattern (see Fig. 4a, b). Due to the nature of the optical lithography and subsequent metal evaporation process, a slight edge of Al is protruding beyond the boundary of the Pd pattern already for the 5 nm deposition of Al prior to any annealing step (inset and white arrows in Fig. 2a).

With an increased thickness of 15 nm, the Al-layer imposes a larger boundary to be overcome by free surface In atoms in order to reach Pd and alloy. In aotms can diffuse back from the Al-layer onto the substrate, assuming a lower diffusion coefficient on Al-oxide than on InAs, and no Al-In alloying occurs. Accordingly, the In concentration on the InAs surface is not substantially changed by the metal, allowing droplet formation everywhere. Slight contrast changes can be detected along the edge of the Pd (see the metal area in Fig. 4b). We attribute this to some In atoms overcoming the Al barrier and diffusing into Pd. We do not observe vertical alloying of the metal into the InAs(111)B substrate as seen on the previous samples (Fig. 2c). The three times thicker Al-layer imposes a boundary for substrate or Pd atoms to penetrate through. Thus, Al acts as a barrier preventing alloying of Pd with free In atoms.

The complete removal of Al is also investigated. For this, circular (diameter 30 µm) and square metal patterns (side length 30 and 15 µm) are fabricated by depositing only 20 nm Pd. After annealing to 600 °C,

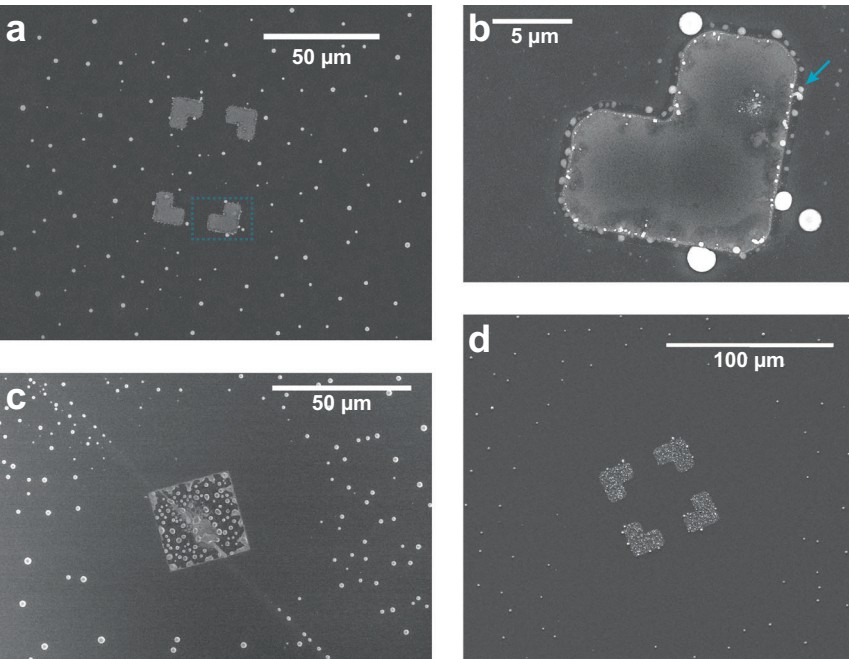

**Fig. 4 | Surface effects due to variations of the metal pattern.** SEM images of **a** metal pattern with increased Al thickness (15 nm) topped by 20 nm Pd inhibiting the formation of the DFZ. **b** Zoom-in of the marked area in (**a**) to highlight the accumulation of In-droplets along the metal rim (arrow). **c** Metal pattern with solely 20 nm Pd exhibiting the DFZ and a more severe vertical alloying into the InAs(111)B substrate. A crystal line defect, diagonally crossing the InAs surface, does not alter the size and shape of the DFZ. **d** New pattern with 5 nm of Al topped by 20 nm Pd making a larger part of the InAs(111)B substrate available for fabrication.

the DFZ emerges once more. Additionally, Fig. 4c shows that the metal pattern alloys almost to the full extent into the InAs(111)B substrate (compare with Supplementary Fig. 8 for 5 nm Al). By fitting an ellipse to the boundary of the DFZ and comparing the ratio between the DFZ and the area of the deposited metal, we observe the same dependency as for the metal stack consisting of 5 nm Al and 20 nm Pd with the DFZ occupying about $84.2 \pm 0.7\%$ of the overall area of interest. Interestingly, the presence of a crystal line defect within the InAs substrate does not appear to alter the DFZ, as can be seen in Fig. 4c. While defects can nucleate and accumulate droplets, we do not observe any additional droplets in the DFZ due to this crystal imperfection. This indicates that the control of In droplet formation via metal patterns is robust towards significant crystal defects in the InAs substrate.

Based on these results, we state that solely the presence of Pd in the metal stack is the driving force for the creation of the DFZ. However, introducing a thin layer of Al can mitigate the otherwise strong alloying of Pd vertically into the InAs substrate. This is relevant for device fabrication, where vertically stacked functional layers are crossing each other. Thicker layers of Al can also suppress the formation of the DFZ, a result of the immiscibility of Al and In and a significant decrease of In diffusion across Al. The presence of Pd will no longer affect free In atoms. Instead, the density of In at the Al boundary increases, leading in the end to droplet formation at the rim.

By adapting a new design for the metal pattern, the DFZ can be increased by $5.6 \pm 0.1\%$ compared to the original pattern. An example can be seen in Fig. 4d, which again exhibits 5 nm Al below 20 nm Pd. This can be attributed to the opening of the inner part of the metal pattern. Therefore, we can conclude that the main alloying of the excess In around the metal only affects the border region. The inner area of the pattern solely alloys with the underlying InAs substrate. Otherwise, we would see a decrease in the size of the DFZ region for the new layout.

Additionally, In-droplets are accumulating on the edge of the metal patch. The new pattern was fabricated via e-beam lithography in contrast to optical lithography used before. This results in a structural

deviation of the edge of the metal stacks and can lead to nucleation points for In-droplets, as well as a larger barrier to overcome, as the width and border of the Al-layer might be different. The effect is an In accumulation at the metal edge discussed above.

## Outlook
Control over droplet formation, a common phenomenon on compound surfaces, is fundamentally interesting and essential for epitaxial processes. We use this to demonstrate the significant possibilities for bottom-up structure formation by geometrically controlling the elemental surface concentration on an InAs surface using metal stacks. The observed effects should be transferable to other semiconductor compounds such as InP, InSb, GaAs or GaN, where droplet formation above the congruent melting temperature is also a naturally occurring phenomena[31,38,52,53]. Metal stacks containing Au, Ti or Pt are commonly employed as electrical contacts on electronic devices and can serve as alloying partners for free In[54–56] or Ga[56–58] surface atoms, respectively. The wide range of metal layering should allow for delicate local tuning of growth parameters by varying composition, material stacking sequence, thickness, and pattern shape. The possibility of designing structures with different functionalities on a single chip is an important feature for more complex devices[5–9]. For example, varying the composition of compound semiconductors enables tuning of the band gap, which is relevant for optical and photovoltaic applications that need to address several wavelength regions simultaneously. Adding a different group V material and annealing, the III-droplets can also serve as nucleation centres for other compound semiconductors. The phenomena observed in this study are robust with regard to different lithography approaches, varying pattern shapes, and the presence of crystal defects. Furthermore, a relation to synthesis processes can be made as the investigated temperature range is suitable for the growth of InAs-based nanowires[39,59], thin films[60] and quantum dots[61]. This study is also important for epitaxy in the presence of prepatterned metal contacts for future more complex growth on top of gate stacks on Si. Thus, our concept opens an important route to combine

lithography definition and synthesis as can be achieved with standard semiconductor fabrication techniques already in use. As the concept is based on a few general assumptions, it should be widely applicable to nano and micro-structured semiconductor synthesis.

## Methods

### Materials

The substrates for all measurements are commercially available n-type InAs(111)B wafer pieces from Wafer Technology. By employing optical and e-beam lithography, we implemented metal stacks in different shapes and sizes (details in the "Results" section) after removing the native InAs oxide via buffered oxide etching. The patterns exhibited different combinations of an Al wetting and an on-top Pd-layer (Al/Pd: 5/20 nm, 15/20 nm, 0/20 nm) after metal evaporation.

### Temperature treatment

To induce In-droplet formation, the samples were heated to 400 °C and exposed to atomic Hydrogen provided by a Hydrogen cracker from MBE Komponenten in UHV to remove the native InAs oxide. Subsequently, the sample temperature was ramped up fast to the desired target value of 500–650 °C (heating rate around 3 °C/s). After reaching the relevant temperature, the heating current was cut off to prevent formed In-droplets from dispersion (cooling rate around 1.5 °C/s). All temperatures were measured with a pyrometer and an emissivity setting of 0.4. The effects described in the paper are observed for each stack, although temperature variations over the sample result in different droplet sizes.

### Analysis techniques

Ex situ measurements were performed in standard imaging mode in a Hitachi SU 8010 SEM. Furthermore, AFM measurements in ambient pressure were done utilising the Nanowizard II from JPK in intermittent contact mode with a highly n-doped Si cantilever (PPP-NCHR from Nanosensors) with a nominal resonance frequency of 330 kHz and a force constant of 42 N/m. To study the formation and movement of In-droplets near the metal edge in situ, an aberration-corrected Spectroscopic PhotoEmission and Low Energy Electron Microscope (SPE-LEEM) connected to the MaxPEEM beamline at MaxIV was employed. Here, a movie was recorded in mirror mode (start voltage −0.1 eV), keeping the sample at a temperature of 550 °C for 24.5 min. To analyse the composition of the area of interest, XPEEM maps of the *In 4d*, *As 3d* and *Al 2p* core level were taken at a photon energy of 100 eV (150 eV for Al 2p). A small cross-section of Pd in the low energy regime did not allow for similar XPEEM measurements for any Pd core level. However, μ-XPS spectra were additionally obtained before and after the annealing at distinct positions on the sample and the metal of the *Pd 3d* (photon energy: 430 eV), *In 4d* (100 eV), *As 3d* (100 eV) and *Al 2p* (200 eV) core level. Reference images were taken in mirror mode and with the Low Energy Electron Microscope.

## Data availability

The XPEEM, XPS, AFM and SEM data generated in this study have been deposited in the Figshare database under https://doi.org/10.6084/m9.figshare.23560146.

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

## Acknowledgements
We acknowledge MAX IV Laboratory for time on Beamline MaxPEEM under Proposal 20200237 (S.B., A.M.). Research conducted at MAX IV, a Swedish national user facility, is supported by the Swedish Research Council under contract 2018-07152, the Swedish Governmental Agency for Innovation Systems under contract 2018-04969, and Formas under contract 2019-02496. J.J. gratefully acknowledges NanoLund for financial support.

## Author contributions
S.B. and A.M. conceived the research plan. S.B., A.J., L.Z. and E.G. performed the experiments, J.J. established and performed the theoretical modelling, S.B. analysed the data, and S.B., J.J. and A.M. wrote the manuscript. S.B., A.J., J.J., L.Z., E.G., L.-E.W. and A.M. contributed to discussions as well as commenting and editing on the manuscript.

## Funding

## Competing interests
The authors declare no competing interests.
