## [Peer Review File · Nature Communications]

REVIEWER COMMENTS

Reviewer #1 (Remarks to the Author):

Benter et al. present a combined experimental and theoretical study demonstrating how Aluminium/Palladium metal patterns can affect the formation of Indium droplets on the surface of InAs(111), when heating the substrate above the congruent evaporation temperatures. The main result of this work is that droplet-free zones can be formed around the lithographically defined Al/Pd structures. SEM, AFM, XPS and XPEEM measurements are combined with theoretical modelling, leading to the conclusion that the Pd layer acts as sink for In atoms. In consequence, the number of diffusing Indium atoms is reduced in the surrounding area, what suppresses the formation of In droplets. Overall, the manuscript provides clear evidence for all its claims, making a consistent story. However, I am not convinced that the present study is suitable for publication in Nature Communications because it will not be of interest to broader audience. Therefore, I recommend publishing in more specialised journal. In my opinion, this work is not of significance to the field and related fields for the following reasons: i) the motivation behind this work seems to be driven by potential applications but the presented metal structures allow only for a very limited droplet-free areas (roughly 2.5 times the size of the metal pattern) which is not very efficient. Authors say that 'the formation of droplets during device manufacturing steps can be also unwanted....' but formation of droplets can be simply prevented by not annealing above the congruent evaporation temperatures; ii) based on rather extensive literature on droplet formation on substrates such as InAs and GaAs one can anticipate that additional materials or even protrusions on the surface will form nucleation sites and can modify the diffusion of atoms. The present work demonstrates the idea but does not advance the field significantly; iii) If I understood correctly, the In clusters are still present in what authors call 'droplet-free zones'. What is essentially achieved therefore is lack of larger clusters/droplets only. Furthermore, the effect is not very robust because as shown in the manuscript, it can only be achieved in rather narrow window of annealing parameters.

Reviewer #2 (Remarks to the Author):

Review of NCOMMS-23-10981: Geometric control of diffusing elements on semiconductor surfaces via stable metal patterns

The manuscript reports an experimental finding where InAs(111)B is heated in vacuum until it reaches incongruent evaporation and the resultant In droplet distribution is controlled by a metal stack. Specifically, a droplet free zone (DFZ) forms around an Al/Pd stack where Pd is believed to act as sinks and Al as a barrier layer. A theoretical model based on a reduced rate equation is provided to quantify the radius of the DFZ. The use of metal stacks as a control method is simple and the results are fundamentally important and likely have practical applications in epitaxy and surface engineering. Though most results are well supported by microscopy and spectroscopy, I find the following issues should be addressed before publication:

1. The manuscript title is inappropriate for two reasons. First, it indicates that the control is effective or present across "semiconductor surfaces". The experiments are specific to InAs(111)B. Would the same results be found on other orientations such as (111)A or (100)? or on other III-V surfaces such as GaAs or InP? There is no discussion or speculation with regard to other surfaces at all. In short, "semiconductor surfaces" in the title could prove an incorrect generalization. Second, "stable metal patterns" in the title is misleading. Since the Al/Pd stack may react and form alloys with the substrate (at least for Al/Pd : 0/20 nm and 5/20 nm stacks), the use of "stable" is chemically incorrect. Further, the boundary of the pattern is irreversibly affected (Fig.2b shows deep etch pits around the boundary after annealing), thus the use of "stable" is geometrically incorrect.

2. Page 7 states that stable In-droplets form with more than eight atoms. Is this (8 atoms) a theoretical projection or experimentally proven fact? Is it applicable at the high annealing temperature of 600 degC in the experiments?

3. Pages 7-8 state that In diffuse back from the Al layer onto InAs surface assuming a lower diffusion coefficient on Al-oxide than on InAs. Where does this "Al-oxide" come from?
4. Concerning the droplet zones (DZ) of Figs. 4a-d, why are the droplets in Fig.4d much smaller than those in Figs.4a-c? What's the annealing temperature of the sample in Fig.4d?
5. The surface of Fig.4c shows a diagonal line. What's the cause of this (tweezer scratch, dislocation)? There must be clean squares elsewhere on the same sample; a different, clean square should be shown in place of this somewhat non-ideal position.
6. Page 10 states the supplier of the cracker as "MBE components", I think the correct name is "MBE Komponenten".

Reviewer #3 (Remarks to the Author):

This is an interesting paper about controlling Indium metal droplet formation using lateral Pd surface sinks instead of commonly used masking or random deposition. The substrate, InAs, surface dissociates with the As evaporating and the In metal diffusing on the surface to accumulate as droplets or be absorbed by patterned Pd islands. Each Pd metal island becomes surrounded by an area denuded of droplets, whose diameter is determined by the kinetics. Al is added underneath the Pd to reduce Pd substrate reaction. It also can act as a lateral barrier to In diffusion into the Pd if too thick.

Noteworthiness: The formation of denuded droplet zones due to surface diffusion into patterned sinks would seem to be something already observed on a compound semiconductor surface. Surface diffusion is imagined in many growth processes particularly of nanowires or epitaxial growth, but usually its from an external flux and ends at a growth interface. There are many examples of metal contacts with reacted radii in the substrate that involve surface segregation and evaporation of one element. The surface surrounding the contact dramatically changes in these cases. There are other examples where a pinhole in an oxide was the sink such as: - the lateral diffusion of deposited Au reacting with Si through random pinholes in its native oxide. A region denuded of Au islands was formed around each pinhole:

Micro-IBA analysis of Au/Si eutectic "crop-circles"

Amato, G; Battiato, A; Vittone, E; Amato, Giampiero; Battiato, Alfio; Croin, Luca; Jaksic, Milko; Siketic, Zdravko; Vignolo, Umberto; Vittone, Ettore

ISSN: 0168-583X , 1872-9584; DOI: 10.1016/j.nimb.2014.10.004

Nuclear instruments & methods in physics research. , 2015, Vol.348, p.183-186

away from novelty is in the application of atomic sink geometries that locally limit metal droplet formations on a surface. The concentration gradient and nucleation rate are a function of temperature and diffusivity that were estimated by a kinetic model. They find a limited temperature range in the case of InAs and Pd sinks which they have investigated.

Significance: I agree that in general, this could be another method to restrict droplet locations for the growth of, for example, quantum dots and wires. And similar strategies could be devised for other compound semiconductor systems. Do the authors have suggestions for sink material other than Pd for In or Ga? They have also avoided significant reaction with the substrate using an Al diffusion barrier.

Support of Conclusions: Yes

Flaws:

I would suggest that they clarify the following:

1. Fig. 1d claims to be plotting the area of the droplet free zone (r_i) versus the area of the metal

(rm). However, the units are clearly not area but radius. Their model equation 1 also uses radii but the area might be more relevant for comparisons of the effects of squares and circles?

2. I am also wondering about the scale bars on the images in figure 1. Are the metal islands all from the same sample shown in (a)? The model plot in 1d shows essentially 3 radii that could be the 2 sizes of circles and the smallest radii about 15 microns could be the squares? Perhaps the scale bar is incorrect in 1a? Also, have they looked at larger Pd islands?

3. The In droplets eventually solidify. Are they faceted and hence single crystalline? The one image in an inset 1d appears to be a droplet without facets? I am wondering about the purity of the droplet after diffusion? Is there remaining As in the In that might induce them to be amorphous or solidify at a different temperature from pure In?

4. The coverage percentage (eg. $5.59 \pm 0.97\%$) would be better written as $(5.6 \pm 1)\%$ since we are not sure whether the error is a % or the entire number is a %.

5. The model line in 1d is based on only 3 radii apparently. The assumption that all parameters in the model are constant would predict a linear line. However, 3 points do not confirm this conclusion. Many different curves can be overlaid through 3 points. I would suggest that the goodness of fit should be evaluated. Can they add error bars to each point? What happens for larger islands? Does it remain linear? I would more likely believe that it will deviate from the linear? Note that there may be an error in the stated units on $F = 1.9 \times 10^{14} \text{ cm}^2 \text{ s}^{-1}$ should be $\# \text{ cm}^{-2} \text{ s}^{-1}$?

6. Methodology fine and standards met.

7. Reproducible yes.

RESPONSE TO REVIEWERS' COMMENTS

Reviewer #1 (Remarks to the Author):

Reviewer:

Benter et al. present a combined experimental and theoretical study demonstrating how Aluminium/Palladium metal patterns can affect the formation of Indium droplets on the surface of InAs(111), when heating the substrate above the congruent evaporation temperatures. The main result of this work is that droplet-free zones can be formed around the lithographically defined Al/Pd structures. SEM, AFM, XPS and XPEEM measurements are combined with theoretical modelling, leading to the conclusion that the Pd layer acts as sink for In atoms. In consequence, the number of diffusing Indium atoms is reduced in the surrounding area, what suppresses the formation of In droplets. Overall, the manuscript provides clear evidence for all its claims, making a consistent story. However, I am not convinced that the present study is suitable for publication in Nature Communications because it will not be of interest to broader audience. Therefore, I recommend publishing in more specialised journal. In my opinion, this work is not of significance to the field and related fields for the following reasons: i) the motivation behind this work seems to be driven by potential applications but the presented metal structures allow only for a very limited droplet-free areas (roughly 2.5 times the size of the metal pattern) which is not very efficient. Authors say that 'the formation of droplets during device manufacturing steps can be also unwanted....' but formation of droplets can be simply prevented by not annealing above the congruent evaporation temperatures; ii) based on rather extensive literature on droplet formation on substrates such as InAs and GaAs one can anticipate that additional materials or even protrusions on the surface will form nucleation sites and can modify the diffusion of atoms. The present work demonstrates the idea but does not advance the field significantly; iii) If I understood correctly, the In clusters are still present in what authors call 'droplet-free zones'. What is essentially achieved therefore is lack of larger clusters/droplets only. Furthermore, the effect is not very robust because as shown in the manuscript, it can only be achieved in rather narrow window of annealing parameters.

Answer:

We are happy that the reviewer agrees that the paper presents a consistent story with clear evidence.

However, we do not agree with the further assessment made by the reviewer and will respond to each of the three points in the following:

(i) the overall size of the droplet-free zone scales with a factor of 6.25 to the size of the metal zone (the radius is 2.5 times larger). This amounts to a very large region on the surface that can be influenced, especially regarding the density of metal patterns on common electronic devices. In device fabrication metal patterns inserted for chip fabrication commonly have areal densities of 30% and larger^{1,2}. Since the droplet free area created by the metal pattern is 6 times larger than the pattern itself, it is evident that with a common chip metal coverage of >30% the effect will be seen on all areas of the chip.

Furthermore, the implemented metal patterns can be tuned in shape and placed deliberately in a specific location on the surface allowing for a very high control over the droplet-free zone. Because these metals are already commonly used as electrical contact material, no additional fabrication tools are necessary guaranteeing an easy implementation into device fabrication. That annealing can always be kept below the non-congruent temperature as stated by the reviewer is also incorrect. It is known for a wide range of semiconductors that increasing the window for annealing is crucial for obtaining high quality defect free crystals and for allowing a variety of growth modes³⁻⁶.

(ii) We agree that there is indeed an extensive literature on droplet formation on III-V compounds. This demonstrates that the field is highly active with continuous work to reach a better understanding on fundamental phenomena and applications. The statement by the reviewer that our study is about the

formation of new nucleation sites via additional materials or protrusions is incorrect. Rather we show that using the available electrical contact material, nucleation of stable droplets can be prohibited by changing the surface pressure of free Indium away from the metal areas.

The concept that we show in our paper has in not been experimentally demonstrated in any of the many previous droplet studies. Even though the effect can perhaps be "anticipated", this has not been the case as no earlier study published and experimentally described it. There is no other experimental study which demonstrates the active mitigation of droplets on the surface of compound semiconductors by implementing metal patterns. Certainly, there is more to do on the subject, but we see our work as a starting point that could lead to many more studies further exploring this effect.

(iii) In clusters in the nm-range are a common phenomenon to occur when heating materials such as InAs, InP or InSb even when removing the native oxide with Hydrogen which is usually done at temperatures around 400°C. Furthermore, the observed effect of the droplet-free zone is quite robust as demonstrated on a variety of samples with different metal composition. Extending the temperature range for fabrication processes without the formation of stable droplets is very important. As soon as the sample temperature exceeds the congruent evaporation temperature, the surface will be supersaturated with element III droplets for III-V compounds. Regarding further processing of the surface, this will induce strong limitations for electrical contacts and characteristics due to the increased surface roughness and the bad interface quality. Furthermore, the implementation of a ternary surface system is strongly limited. The new component will predominantly interact with the droplet and not the surface⁷. By removing the excess elemental concentration very different growth regimes become available which is important for tuning crystal structure, quality and composition.

We also disagree with the statement that a narrow temperature window is not important. A variety of studies have e.g. clearly demonstrated that an increase of the temperature window during the growth of as little as 30°C can result in very different crystal structure and quality^{3-6,8}.

Reviewer #2 (Remarks to the Author):

Reviewer:

Review of NCOMMS-23-10981: Geometric control of diffusing elements on semiconductor surfaces via stable metal patterns

The manuscript reports an experimental finding where InAs(111)B is heated in vacuum until it reaches incongruent evaporation and the resultant In droplet distribution is controlled by a metal stack. Specifically, a droplet free zone (DFZ) forms around an Al/Pd stack where Pd is believed to act as sinks and Al as a barrier layer. A theoretical model based on a reduced rate equation is provided to quantify the radius of the DFZ. The use of metal stacks as a control method is simple and the results are fundamentally important and likely have practical applications in epitaxy and surface engineering. Though most results are well supported by microscopy and spectroscopy, I find the following issues should be addressed before publication:

- 1. The manuscript title is inappropriate for two reasons. First, it indicates that the control is effective or present across "semiconductor surfaces". The experiments are specific to InAs(111)B. Would the same results be found on other orientations such as (111)A or (100)? or on other III-V surfaces such as GaAs or InP? There is no discussion or speculation with regard to other surfaces at all. In short, "semiconductor surfaces" in the title could prove an incorrect generalization. Second, "stable metal patterns" in the title is misleading. Since the Al/Pd stack may react and form alloys with the substrate*

(at least for Al/Pd : 0/20 nm and 5/20 nm stacks), the use of "stable" is chemically incorrect. Further, the boundary of the pattern is irreversibly affected (Fig.2b shows deep etch pits around the boundary after annealing), thus the use of "stable" is geometrically incorrect.

Answer:

We follow the reviewers concern about the chosen title. As the observed behaviour can be understood based on a few general assumptions, the concept should be applicable across a wide range of compound semiconductors. However, we show it experimentally for InAs, and it is good to be clear about that in the title and have thus adapted this suggestion. We changed the title to [Geometric control of diffusing elements on InAs semiconductor surfaces via metal patterns] to make it more precise and included additional discussion of the wider applicability.

Further, we have expanded the discussion in the paper on the applicability of the results for other compound semiconductors and a variety of surfaces. The formation of droplets is found on a wide variety of compound semiconductors, including InAs, InP, GaAs, GaP, GaSb, GaN, InSb⁹⁻¹⁴. In regard to crystal facets, droplets are also observed on a variety of facets of III-V semiconductors. (111A: ¹⁵⁻¹⁷, 100: ^{18,19}, 110: ^{20,21}). Deposition of metal islands as in the present case is possible on all these facets. Diffusion coefficients will depend on crystal facet and material, which again will change temperature windows. But the concept should be widely applicable. With regard to the specific metal stack used, the Al/Pd should work for the In containing systems such as InAs, InP and InSb. For compound semiconductors including Ga the chosen metal composition would be feasible to allow for Ga diffusion and alloying into the Pd²². In general, if one metal within the lithography stack can alloy with the element forming droplets on the surface, the described effect in the paper should be visible.

Reviewer:

2. Page 7 states that stable In-droplets form with more than eight atoms. Is this (8 atoms) a theoretical projection or experimentally proven fact? Is it applicable at the high annealing temperature of 600 degC in the experiments?

Answer:

We thank the reviewer for pointing this out. The number of atoms considered to form a stable cluster are a prediction, however the number 8 is a typo. For our experiments, we assume a critical size of 39 as stated correctly in the SI. We have removed the sentence in question since it was not bringing more clarity to the manuscript, and it is enough to state the critical size in the SI together with the other parameters. Some parameters of the reduced rate equation approach can be chosen more or less freely including the number of In atoms (i) that form stable droplets. Based on experimental observations for annealing to 600°C, we determined $i=39$ in agreement with the droplet density observed far away from the metal stack. At higher annealing temperature, we would expect a smaller critical size due to the higher supersaturation of In atoms in this case as an effect of the faster release of In atoms from the surface. This is explained in the SI and manuscript.

Reviewer:

3. Pages 7-8 state that In diffuse back from the Al layer onto InAs surface assuming a lower diffusion coefficient on Al-oxide than on InAs. Where does this "Al-oxide" come from?

Answer:

After lift-off of the photoresist (the final step of the metal stack preparation), all samples are exposed

to ambient conditions during transfer into ultra-high vacuum. The part of the stack exposed to air will form a stable native surface oxide which for Al will be ~ 3 nm²³. E.g. for the sample with a 15 nm thick Al layer, the Al is protruding slightly beyond the edges of the Pd on top. This part will react with the oxygen in the atmosphere resulting in a stable layer of Al-oxide at the surface of ~ 3 nm. And generally, any exposed Al will have such a thin oxide.

Reviewer:

4. Concerning the droplet zones (DZ) of Figs. 4a-d, why are the droplets in Fig.4d much smaller than those in Figs.4a-c? What's the annealing temperature of the sample in Fig.4d?

Answer:

The sample in figure 4d was annealed to a temperature of around 600°C corresponding to a similar temperature as for the samples shown in figure 4a-c. However, the field of view varies between a, c and d (a: 186 μ m x 145 μ m; c: 156 μ m x 108 μ m; d: 241 μ m x 216 μ m). Therefore, the In droplets appear smaller in d. The average size of the In droplets displayed is roughly 1.3 μ m², which is within the range of other samples. In order to provide more clarity for the reader, we improved the figure to better show this difference.

Reviewer:

5. The surface of Fig.4c shows a diagonal line. What's the cause of this (tweezer scratch, dislocation)? There must be clean squares elsewhere on the same sample; a different, clean square should be shown in place of this somewhat non-ideal position.

Answer:

This comment of the reviewer led to significant discussion and consideration on our side about the use of this image in the paper. The cause of the 'diagonal line' is a crystal line dislocation in the substrate itself and after careful consideration we finally concluded it to be important to showcase that such a natural imperfection within the sample does not alter the formation of the droplet-free zone as can be seen in figure R1. Here we have compared it to images of lithographic patterns with no dislocation across the metal plate. Production and manufacturing techniques are tailored to provide as close to ideal surfaces as possible. However, a chance of imperfections remains. Kanjanachuchi et. al²⁴ showed that a crystal dislocation on the InAs surface can accumulate running droplets. In our study, we demonstrate that this does not influence the evolution of the droplet-free zone. As can be seen in figure R1, no additional droplets within the droplet-free zone are generated independent of the origin of the defect. We now explicitly mention this in the manuscript as we think it improves the work and thank the reviewer for pointing this out.

Figure R1: Pattern with 5 nm of Al topped by 20 nm Pd on an InAs(111)B substrate after removing the native oxide and annealing to 600°C without (a) and with defects in the substrate surface (b). Pattern with solely 20 nm of Pd with (c) and without (d) defects in the surface. It is seen that inhomogeneities in the substrate do not influence the formation of the droplet-free zone important for large-scale fabrication, but rather agglomerate droplet when positioned outside of the DFZ.

Reviewer:

6. Page 10 states the supplier of the cracker as "MBE components", I think the correct name is "MBE Komponenten".

Answer:

We thank the reviewer for this correction and implemented it in the manuscript.

Reviewer #3 (Remarks to the Author):

Reviewer:

This is an interesting paper about controlling Indium metal droplet formation using lateral Pd surface sinks instead of commonly used masking or random deposition. The substrate, InAs, surface dissociates with the As evaporating and the In metal diffusing on the surface to accumulate as droplets or be absorbed by patterned Pd islands. Each Pd metal island becomes surrounded by an area denuded of droplets, whose diameter is determined by the kinetics. Al is added underneath the Pd to reduce Pd substrate reaction. It also can act as a lateral barrier to In diffusion into the Pd if too thick.

Noteworthiness: The formation of denuded droplet zones due to surface diffusion into patterned sinks would seem to be something already observed on a compound semiconductor surface. Surface diffusion is imagined in many growth processes particularly of nanowires or epitaxial growth, but usually its from an external flux and ends at a growth interface. There are many examples of metal contacts with reacted radii in the substrate that involve surface segregation and evaporation of one element. The surface surrounding the contact dramatically changes in these cases. There are other examples where a pinhole in an oxide was the sink such as: - the lateral diffusion of deposited Au reacting with Si through random pinholes in its native oxide. A region denuded of Au islands was formed around each pinhole:

Micro-IBA analysis of Au/Si eutectic "crop-circles"

Amato, G; Battiato, A; Vittone, E; Amato, Giampiero; Battiato, Alfio; Croin, Luca; Jaksic, Milko; Siketic, Zdravko; Vignolo, Umberto; Vittone, Ettore

ISSN: 0168-583X , 1872-9584; DOI: 10.1016/j.nimb.2014.10.004

Nuclear instruments & methods in physics research. , 2015, Vol.348, p.183-186

away from novelty is in the application of atomic sink geometries that locally limit metal droplet formations on a surface. The concentration gradient and nucleation rate are a function of temperature and diffusivity that were estimated by a kinetic model. They find a limited temperature range in the case of InAs and Pd sinks which they have investigated.

Answer:

We thank the reviewer for pointing to this reference which is really interesting, and we have included it as a good addition to our introduction of the subject in the paper.

Significance: I agree that in general, this could be another method to restrict droplet locations for the growth of, for example, quantum dots and wires. And similar strategies could be devised for other compound semiconductor systems. Do the authors have suggestions for sink material other than Pd for In or Ga? They have also avoided significant reaction with the substrate using an Al diffusion barrier.

Answer:

In principle, any metal that allows for alloying with In or Ga is a suitable candidate. Based on the individual phase diagrams, we think that Au^{25,26}, Ti^{27,28} or Pt²⁹ would also facilitate a droplet free zone. All phase diagrams, indicate a variety of possible alloys between the substrate element and each metal for annealing temperatures above the congruent melting temperature of Ga- or In-V compounds. This corresponds well with the In-Pd phase diagram³⁰, fundamental to the evolution of the droplet-free zone on InAs, which also displays a variety of potential alloys depending on the amount of In diffusing into the metal stack. We thank the reviewer for the encouraging and insightful comments and adapted the manuscript with the new information.

Support of Conclusions: Yes

Reviewer:

Flaws:

I would suggest that they clarify the following:

1. Fig. 1d claims to be plotting the area of the droplet free zone (r_i) versus the area of the metal (r_m). However, the units are clearly not area but radius. Their model equation 1 also uses radii but the area might be more relevant for comparisons of the effects of squares and circles?

Answer:

We thank the reviewer for pointing this out and clarified in the manuscript that the plot shows indeed the dependency on the radii of the metal vs the droplet-free zone. We agree in principle with the reviewer's statement regarding the relevance of the area compared to the radius and have given this considerable consideration. However, the metal stacks in our study are in general very compact (circle and square). Therefore, we believe that by working with the radius instead of the overall area the size distribution and overall dependency is more accessible for the reader and emphasizes the connection to the theoretical model in a clear way. For more complicated structures with a side length ratio of 1:n (with $n > 1$), a comparison of the different areas would be more relevant. In the paper we now clarify the use of area and radius.

Reviewer:

2. I am also wondering about the scale bars on the images in figure 1. Are the metal islands all from the same sample shown in (a)? The model plot in 1d shows essentially 3 radii that could be the 2 sizes of circles and the smallest radii about 15 microns could be the squares? Perhaps the scale bar is incorrect in 1a? Also, have they looked at larger Pd islands?

Answer:

We thank the reviewer for this important comment. The scale bar in figure 1a should be 250 μm instead of 50 μm . We corrected the figure accordingly. In general, larger metal patterns were only implemented for samples exhibiting the 15 nm thick Al layer inhibiting the formation of the droplet-free zone and therefore not relevant for the plot in figure 1d. We checked all figures in the main text and SI to make sure that the correct details are implemented.

Reviewer:

3. The In droplets eventually solidify. Are they faceted and hence single crystalline? The one image in an inset 1d appears to be a droplet without facets? I am wondering about the purity of the droplet after diffusion? Is there remaining As in the In that might induce them to be amorphous or solidify at a different temperature from pure In?

Answer:

This is a very interesting question. Our XPEEM studies did not indicate any As present in the In droplets as shown in the supplementary information. In vacuum we do not expect a significant As content in the droplet as the As will evaporate. However, if an external As pressure is added InAs formation can occur^{21,31,32}. It is possible that during cooling some As from the surface could enter the droplet, but we have seen no signs of this. We agree that there are no pronounced facets on the droplets visible with SEM. However, the footprint of droplets close to the metal pattern indicates that there are facets present at room temperature, see figure R2. We did not investigate the characteristics of the In droplets further since bulk Indium has a melting point of 156°C, so significantly lower than the annealing temperatures in the present study.

Figure R2: Footprint of an In droplet close to a metal pattern after annealing to 600°C and cooling down. The excess In is alloying into the Pd during cooling down of the sample leaving only a small droplet behind (white protrusion).

Reviewer:

4. The coverage percentage (eg. $5.59 \pm 0.97\%$) would be better written as $(5.6 \pm 1)\%$ since we are not sure whether the error is a % or the entire number is a %.

Answer:

We thank the reviewer for this comment and changed the text accordingly.

Reviewer:

5. The model line in 1d is based on only 3 radii apparently. The assumption that all parameters in the model are constant would predict a linear line. However, 3 points do not confirm this conclusion. Many different curves can be overlaid through 3 points. I would suggest that the goodness of fit should be evaluated. Can they add error bars to each point? What happens for larger islands? Does it remain linear? I would more likely believe that it will deviate from the linear? Note that there may be an error in the stated units on $F = 1.9 \times 10^{14} \text{ cm}^2\text{s}^{-1}$ should be $\# \text{ cm}^{-2}/\text{s}^{-1}$?

Answer:

We agree that there is an error in the unit for the decomposition rate and changed it accordingly. We understand the concern of the reviewer about the goodness of the fit and determine $R^2=0.973$ for the fit shown in figure 1d. It is relevant to note, that the fit is forced through the origin, since a pure InAs sample without any metal would not yield any droplet-free zone. We implemented error bars in figure 1d.

We agree that at sufficiently large patterns, the size of the droplet-free zone might not scale linear with the size of the metal pattern. In general, there are two limiting factors for the size of the droplet-free zone: (i) the amount of sink material available, ergo the size of the metal stack, and (ii) the diffusion coefficient of the droplet-free zone. Depending on the size of the metal, one factor is more dominant than the other. Given that the diffusion coefficient of In atoms on InAs surface is in the range of $10^3 \mu\text{m}^2\text{s}^{-1}$ and our process takes several minutes, In atoms originating from the surface several millimetre away could reach the metal pattern in principle. Therefore, the limiting factor is the amount

of alloying partner for the In atoms (= the metal size). We agree that eventually the size of the droplet-free zone might not scale linear with the size of the metal pattern, reaching the point where (ii) is imposing stronger restrictions resulting in an overall logarithmic fitting function. However, the investigated pattern size range in this study is located in the onset of this logarithmic fit and can hence be estimated with a linear fit function.

Reviewer:

6. Methodology fine and standards met.

7. Reproducible yes.

1. Kahng, A. B. & Topaloglu, R. O. Performance-Aware CMP Fill Pattern Optimization. in (2007).
2. Stine, B. E. *et al.* The physical and electrical effects of metal-fill patterning practices for oxide chemical-mechanical polishing processes. *IEEE Trans. Electron Devices* **45**, 665–679 (1998).
3. Sritirawisarn, N. & Nötzel, R. InAs/InP Quantum Dots, Dashes, and Ordered Arrays. *Jpn. J. Appl. Phys.* **48**, (2009).
4. LaBella, V. P., Krause, M. R., Ding, Z. & Thibado, P. M. Arsenic-rich GaAs(0 0 1) surface structure. *Surf. Sci. Rep.* **60**, 1–53 (2005).
5. Caroff, P. *et al.* Controlled polytypic and twin-plane superlattices in III-V nanowires. *Nat. Nanotechnol.* **4**, 50–55 (2008).
6. Koleske, D. D., Wickenden, A. E., Henry, R. L., DeSisto, W. J. & Gorman, R. J. Growth model for GaN with comparison to structural, optical, and electrical properties. *J. Appl. Phys.* **84**, 1998–2010 (1998).
7. Wang, L. *et al.* Novel Dilute Bismide, Epitaxy, Physical Properties and Device Application. *Crystals* **7**, 63 (2017).
8. Arciprete, F. *et al.* Temperature dependence of the size distribution function of InAs quantum dots on GaAs(001). *Phys. Rev. B* **81**, 165306 (2010).
9. Shusterman, S., Paltiel, Y., Sher, A., Ezersky, V. & Rosenwaks, Y. High-density nanometer-scale InSb dots formation using droplets heteroepitaxial growth by MOVPE. *J. Cryst. Growth* **291**, 363–369 (2006).
10. Genova, F., Papuzza, C., Rigo, C. & Stano, S. Effect of InP substrate thermal degradation on MBE InGaAs layers. *J. Cryst. Growth* **69**, 635–638 (1984).
11. AbuWaar, Z. Y., Wang, Z. M., Lee, J. H. & Salamo, G. J. Observation of Ga droplet formation on (311)A and (511)A GaAs surfaces. *Nanotechnology* **17**, 4037–4040 (2006).
12. Lugstein, A. *et al.* Study of focused ion beam response of GaSb. *Nucl. Instruments Methods Phys. Res. Sect. B Beam Interact. with Mater. Atoms* **255**, 309–313 (2007).
13. Karpov, S. Y., Bord, O. V., Talalaev, R. A. & Makarov, Y. N. Gallium droplet formation during MOVPE and thermal annealing of GaN. *Mater. Sci. Eng. B Solid-State Mater. Adv. Technol.* **82**, 22–24 (2001).
14. Park, S. *et al.* Effects of TMSb overpressure on InSb surface morphology for InSb epitaxial

- growth using low pressure metalorganic chemical vapor deposition. *J. Cryst. Growth* **401**, 518–522 (2014).
15. Bietti, S. *et al.* High-temperature droplet epitaxy of symmetric GaAs/AlGaAs quantum dots. *Sci. Rep.* **10**, 1–10 (2020).
 16. Tuktamyshev, A. *et al.* Nucleation of Ga droplets self-assembly on GaAs(111)A substrates. *Sci. Rep.* **11**, 1–11 (2021).
 17. Kanjanachuchai, S. & Euaruksakul, C. Self-Running Ga Droplets on GaAs (111)A and (111)B Surfaces. *ACS Appl. Mater. Interfaces* **5**, 7709–7713 (2013).
 18. Bietti, S. *et al.* Precise shape engineering of epitaxial quantum dots by growth kinetics. *Phys. Rev. B* **92**, 075425 (2015).
 19. Tersoff, J., Jesson, D. E. & Tang, W. X. Running Droplets of Gallium from Evaporation of Gallium Arsenide. *Science (80-.)*. **324**, (2009).
 20. Mattila, M., Hakkarainen, T., Jiang, H., Kauppinen, E. I. & Lipsanen, H. Effect of substrate orientation on the catalyst-free growth of InP nanowires. *Nanotechnology* **18**, 155301 (2007).
 21. Potts, H., Morgan, N. P., Tütüncüoğlu, G., Friedl, M. & Morral, A. F. I. Tuning growth direction of catalyst-free InAs(Sb) nanowires with indium droplets. *Nanotechnology* **28**, (2017).
 22. Schwerin, J., Müller, D., Kiese, S. & Gille, P. Single crystal growth in the Ga–Pd system. *J. Cryst. Growth* **401**, 613–616 (2014).
 23. Evertsson, J. *et al.* The thickness of native oxides on aluminum alloys and single crystals. *Appl. Surf. Sci.* **349**, 826–832 (2015).
 24. Kanjanachuchai, S. & Photongkam, P. Dislocation-Guided Self-Running Droplets. *Cryst. Growth Des.* **15**, 14–19 (2015).
 25. Liu, H. S., Cui, Y., Ishida, K. & Jin, Z. P. Thermodynamic reassessment of the Au In binary system. *Calphad* **27**, 27–37 (2003).
 26. Okamoto, H. Au-Ga (Gold-Gallium). *J. Phase Equilibria Diffus.* **34**, 174–175 (2013).
 27. Gulay, L. D. & Schuster, J. C. Investigation of the titanium–indium system. *J. Alloys Compd.* **360**, 137–142 (2003).
 28. Okamoto, H. Ga-Ti (Gallium-Titanium). *J. Phase Equilibria Diffus.* **26**, 398 (2005).
 29. Guex, P. & Feschotte, P. Les systèmes binaires platine-aluminium, platinegallium et platine-indium. *J. Less Common Met.* **46**, 101–116 (1976).
 30. Okamoto, H. In-Pd (Indium-Palladium). *J. Phase Equilibria* **24**, 481 (2003).
 31. Lee, J. H., Wang, Z. M. & Salamo, G. J. The Control on Size and Density of InAs QDs by Droplet Epitaxy. *IEEE Trans. Nanotechnol.* **8**, 431–436 (2009).
 32. Grap, T. *et al.* Self-catalyzed VLS grown InAs nanowires with twinning superlattices. *Nanotechnology* **24**, 335601 (2013).

REVIEWERS' COMMENTS

Reviewer #1 (Remarks to the Author):

I would like to thank authors for their replies. I have interleaved my comments/replies below: We are happy that the reviewer agrees that the paper presents a consistent story with clear evidence. However, we do not agree with the further assessment made by the reviewer and will respond to each of the three points in the following:

(i) the overall size of the droplet-free zone scales with a factor of 6.25 to the size of the metal zone (the radius is 2.5 times larger). This amounts to a very large region on the surface that can be influenced, especially regarding the density of metal patterns on common electronic devices. In device fabrication metal patterns inserted for chip fabrication commonly have areal densities of 30% and larger [1,2]. Since the droplet free area created by the metal pattern is 6 times larger than the pattern itself, it is evident that with a common chip metal coverage of >30% the effect will be seen on all areas of the chip. Furthermore, the implemented metal patterns can be tuned in shape and placed deliberately in a specific location on the surface allowing for a very high control over the droplet-free zone. Because these metals are already commonly used as electrical contact material, no additional fabrication tools are necessary guaranteeing an easy implementation into device fabrication.

It would be useful to indicate what 'common electronic devices' authors have in mind. With semiconductor manufacturing technologies being scaled down to nanometre size, it is not clear at what stage of the manufacturing process the investigated here structures, with micron sized metal patterns, could be useful.

That annealing can always be kept below the non-congruent temperature as stated by the reviewer is also incorrect. It is known for a wide range of semiconductors that increasing the window for annealing is crucial for obtaining high quality defect free crystals and for allowing a variety of growth modes (3–6).

I did not say that the annealing 'can always be kept below the non-congruent temperature', but that 'formation of droplets can be simply prevented by not annealing above the congruent evaporation temperatures'. The references (3-5) cited by authors above refer to growth conditions supporting formation and growth of nanostructures such as quantum dots, nanowires etc., instead of clean crystal surfaces, and do not support the statement that annealing above the congruent temperature is required for obtaining high quality defect free crystals.

(ii) We agree that there is indeed an extensive literature on droplet formation on III-V compounds. This demonstrates that the field is highly active with continuous work to reach a better understanding on fundamental phenomena and applications. The statement by the reviewer that our study is about the formation of new nucleation sites via additional materials or protrusions is incorrect. Rather we show that using the available electrical contact material, nucleation of stable droplets can be prohibited by changing the surface pressure of free Indium away from the metal areas.

It seems like the authors like to take phrases out of context in order to support their statement and to highlight that the reviewer is incorrect. As I have previously said 'based on rather extensive literature on droplet formation on substrates such as InAs and GaAs one can anticipate that additional materials or even protrusions on the surface will form nucleation sites and can modify the diffusion of atoms.' This refers to the main mechanism behind the results in this manuscript, mainly that 'Pd-layer acts as sink for the free In atoms. As a result, the overall In-density around the metal patterns is lowered impeding the formation of droplets. Fig.1a depicts the concept.' If the authors disagree with the above statement about nucleation site and modified diffusion, they should perhaps rewrite the manuscript, because it contradicts what is written in the abstract and the main body of the text.

The concept that we show in our paper has in not been experimentally demonstrated in any of the many previous droplet studies. Even though the effect can perhaps be "anticipated", this has not been the case as no earlier study published and experimentally described it. There is no other

experimental study which demonstrates the active mitigation of droplets on the surface of compound semiconductors by implementing metal patterns. Certainly, there is more to do on the subject, but we see our work as a starting point that could lead to many more studies further exploring this effect.

Agreed. Also, thank you for agreeing that the effect could be anticipated based on previous studies. 

(iii) In clusters in the nm-range are a common phenomenon to occur when heating materials such as InAs, InP or InSb even when removing the native oxide with Hydrogen which is usually done at temperatures around 400°C. Furthermore, the observed effect of the droplet-free zone is quite robust as demonstrated on a variety of samples with different metal composition. Extending the temperature range for fabrication processes without the formation of stable droplets is very important. As soon as the sample temperature exceeds the congruent evaporation temperature, the surface will be supersaturated with element III droplets for III-V compounds. Regarding further processing of the surface, this will induce strong limitations for electrical contacts and characteristics due to the increased surface roughness and the bad interface quality. Furthermore, the implementation of a ternary surface system is strongly limited. The new component will predominantly interact with the droplet and not the surface 7. By removing the excess elemental concentration very different growth regimes become available which is important for tuning crystal structure, quality and composition.

Agreed.

We also disagree with the statement that a narrow temperature window is not important. A variety of studies have e.g. clearly demonstrated that an increase of the temperature window during the growth of as little as 30°C can result in very different crystal structure and quality

The authors again take phrases out of context in order to support their statements. What was pointed out is that the effect is not very robust because it can only be achieved in rather narrow window of annealing parameters. And therefore 'this parameter window' will not necessary overlap with the parameters required during the manufacturing process or other application process. It has nothing to do with obvious statement that 'as little as 30deg can result in very different crystal structure and quality'. The authors reply is not adequate.

Reviewer #2 (Remarks to the Author):

Thank you for clarifying and modifying the title and content of the previous version. I found the rebuttal convincing and the changes made satisfactory. I have no further issues and recommend the manuscript be accepted in the present form.

Reviewer #3 (Remarks to the Author):

They have addressed all of my concerns well.

RESPONSE TO REVIEWERS' COMMENTS

Reviewer #1 (Remarks to the Author):

I would like to thank authors for their replies. I have interleaved my comments/replies below:

Answer 1st re-submission:

We are happy that the reviewer agrees that the paper presents a consistent story with clear evidence. However, we do not agree with the further assessment made by the reviewer and will respond to each of the three points in the following:

(i) the overall size of the droplet-free zone scales with a factor of 6.25 to the size of the metal zone (the radius is 2.5 times larger). This amounts to a very large region on the surface that can be influenced, especially regarding the density of metal patterns on common electronic devices. In device fabrication metal patterns inserted for chip fabrication commonly have areal densities of 30% and larger^{1,2}. Since the droplet free area created by the metal pattern is 6 times larger than the pattern itself, it is evident that with a common chip metal coverage of >30% the effect will be seen on all areas of the chip. Furthermore, the implemented metal patterns can be tuned in shape and placed deliberately in a specific location on the surface allowing for a very high control over the droplet-free zone. Because these metals are already commonly used as electrical contact material, no additional fabrication tools are necessary guaranteeing an easy implementation into device fabrication.

Reviewer #1:

It would be useful to indicate what 'common electronic devices' authors have in mind. With semiconductor manufacturing technologies being scaled down to nanometre size, it is not clear at what stage of the manufacturing process the investigated here structures, with micron sized metal patterns, could be useful.

Answer 2nd submission:

We thank the reviewer for the suggestion. The scaling down of the semiconductor manufacturing to the nanometer level is pertains to structural features of e.g. individual transistors on a chip. However, the overall chip layout imbedding these structures is hierarchical including features of various length scales up to many micrometers. Thus, the ability to change growth conditions in different areas on the micrometer scale is relevant to allow different functionalities across a chip. The use of compound semiconductors (benefitting from this study) in such larger scale structures has been found in applications like CCD and pixel arrays¹, image sensors² and radio antennas going back quite some years. More recently advances towards wearable antennas³, monolithic integration of III-V optoelectronics on Si⁴ and on-chip NIR spectrometers used for a wide range of applications from industry to agriculture^{5,6}. We adapted the manuscript accordingly.

Answer 1st re-submission:

That annealing can always be kept below the non-congruent temperature as stated by the reviewer is also incorrect. It is known for a wide range of semiconductors that increasing the window for annealing is crucial for obtaining high quality defect free crystals and for allowing a variety of growth modes (3–6).

Reviewer #1:

I did not say that the annealing 'can always be kept below the non-congruent temperature', but that 'formation of droplets can be simply prevented by not annealing above the congruent evaporation temperatures'. The references (3-5) cited by authors above refer to growth conditions supporting formation and growth of nanostructures such as quantum dots, nanowires etc., instead of clean crystal surfaces, and do not support the statement that annealing above the congruent temperature is required for obtaining high quality defect free crystals.

Answer 2nd submission:

To further elaborate on the final part of our answer to the first comments of the reviewer:

The statement '*formation of droplets can be simply prevented by not annealing above the congruent evaporation temperatures*' made by the reviewer implies that as long as droplets are undesirable on the surface of the compound semiconductor and no further measures are taken to avoid droplet formation, the process temperature must always be kept below the non-congruent temperature. We would like to point out that increased annealing temperatures can facilitate desirable characteristics such as high-quality growth in compound semiconductors as mentioned before (see also reference in next paragraph) although we do not elaborate this point in the manuscript.

Regarding the statement "[...] high quality defect free crystals and for allowing a variety of growth modes [...]" (see above), reference 3 (Sritirawisarn et al.) and 4 (LaBela et al.) refer to the variety of possible growth modes available by increasing the process temperature window, as stated in the 1st answer. Reference 5 (Caroff et al.) manufactured wurtzite nanowires with a defect density of below 1% of the total wire length, i.e. defect free crystals. Indeed, we are commenting here on nanocrystals since this is a prominent and recent example on the importance of extending and using temperature windows for crystal control. The optimal growth windows for III-Vs are often above the congruent temperature overlapping with the temperature range explored in the manuscript ⁷.

Answer 1st re-submission:

(ii) We agree that there is indeed an extensive literature on droplet formation on III-V compounds. This demonstrates that the field is highly active with continuous work to reach a better understanding on fundamental phenomena and applications. The statement by the reviewer that our study is about the formation of new nucleation sites via additional materials or protrusions is incorrect. Rather we show that using the available electrical contact material, nucleation of stable droplets can be prohibited by changing the surface pressure of free Indium away from the metal areas.

Reviewer #1:

It seems like the authors like to take phrases out of context in order to support their statement and to highlight that the reviewer is incorrect. As I have previously said ' based on rather extensive literature on droplet formation on substrates such as InAs and GaAs one can anticipate that additional materials or even protrusions on the surface will form nucleation sites and can modify the diffusion of atoms.' This refers to the main mechanism behind the results in this manuscript, mainly that 'Pd-layer acts as sink for the free In atoms. As a result, the overall In-density around the metal patterns is lowered impeding the formation of droplets. Fig. 1a depicts the concept.' If the authors disagree with the above statement about nucleation site and modified diffusion, they should perhaps rewrite the manuscript, because it contradicts what is written in the abstract and the main body of the text.

Answer 2nd submission:

We have no intention of taking phrases out of context in order to support statements or to highlight that the reviewer is incorrect, as claimed by the reviewer. And we find no support for this claim in the reviewer comments. We again note as we did in our previous answer that in previous literature the concepts that we develop in our paper were not anticipated or developed. If our results and concepts could be anticipated (or not) is not really relevant as it has not been done and explored in published form. Further we again find it important to clarify that 'nucleation sites' formed by the structures as mentioned again by the reviewer is not a part of the mechanism of the Pd stacks described in the manuscript as stated in the manuscript and previous answer.

Answer 1st re-submission:

The concept that we show in our paper has in not been experimentally demonstrated in any of the many previous droplet studies. Even though the effect can perhaps be "anticipated", this has not been the case as no earlier study published and experimentally described it. There is no other experimental study which demonstrates the active mitigation of droplets on the surface of compound semiconductors by implementing metal patterns. Certainly, there is more to do on the subject, but we see our work as a starting point that could lead to many more studies further exploring this effect.

Reviewer #1:

Agreed. Also, thank you for agreeing that the effect could be anticipated based on previous studies.

Answer 1st re-submission:

(iii) In clusters in the nm-range are a common phenomenon to occur when heating materials such as InAs, InP or InSb even when removing the native oxide with Hydrogen which is usually done at temperatures around 400°C. Furthermore, the observed effect of the droplet-free zone is quite robust as demonstrated on a variety of samples with different metal composition. Extending the temperature range for fabrication processes without the formation of stable droplets is very important. As soon as the sample temperature exceeds the congruent evaporation temperature, the surface will be supersaturated with element III droplets for III-V compounds. Regarding further processing of the surface, this will induce strong limitations for electrical contacts and characteristics due to the increased surface roughness and the bad interface quality. Furthermore, the implementation of a ternary surface system is strongly limited. The new component will predominantly interact with the droplet and not the surface. By removing the excess elemental concentration very different growth regimes become available which is important for tuning crystal structure, quality and composition.

Reviewer #1:

Agreed.

Answer 1st re-submission:

We also disagree with the statement that a narrow temperature window is not important. A variety of studies have e.g. clearly demonstrated that an increase of the temperature window during the growth of as little as 30°C can result in very different crystal structure and quality

Reviewer #1:

The authors again take phrases out of context in order to support their statements. What was pointed out is that the effect is not very robust because it can only be achieved in rather narrow window of annealing parameters. And therefore 'this parameter window' will not necessary overlap with the parameters required during the manufacturing process or other application process. It has nothing to do with obvious statement that 'as little as 30deg can result in very different crystal structure and quality'. The authors reply is not adequate.

Answer 2nd submission:

As stated before, extensions of the available temperature can have a dramatic result on e.g. crystal structure or quality. Apparently, the reviewer agrees that a 30°C annealing temperature window is relevant (even obviously so). Our study enables an additional range of a minimum of 50°C depending on the process speed. The temperatures

covered in this study are also within the range of III-V compound synthesis including InAs (see for example ^{8,9} for nanowires, ^{10,11} for thin films and quantum dots).

Furthermore, the limitations in terms of temperature and time are already discussed in section 'Indium droplet formation on InAs near metal patterns as a function of temperature' of the manuscript, as well as in Supplementary Note 1 and 7. We have discussed the robustness in terms of using different methods to print the metal patterns (it works for both electron and optical lithography) as well as different shapes of the metal stack. Further, we added in the previous revision that it also works in the presence of crystal defects.

Reviewer #2:

Thank you for clarifying and modifying the title and content of the previous version. I found the rebuttal convincing and the changes made satisfactory. I have no further issues and recommend the manuscript be accepted in the present form.

Reviewer #3:

They have addressed all of my concerns well.

References:

1. Holland, A. D. New developments in CCD and pixel arrays. *Nucl. Instruments Methods Phys. Res. Sect. A Accel. Spectrometers, Detect. Assoc. Equip.* **513**, 308–312 (2003).
2. C.L. Chen, D-R. Yost, J.M. Knecht, D.C. Chapman, D.C. Oakley, L. J. M., J.P. Donnelly, A.M. Soares, V. Suntharalingam, R. Berger, V. Bolkhovsky, W. H. & B.D. Wheeler, C.L. Keast, and D. C. S. Wafer-Scale 3D Integration of InGaAs Image Sensors with Si Readout Circuits. in *2009 IEEE International Conference on 3D System Integration* (2009). doi:10.1109/3DIC.2009.5306556.
3. Yeongin Kim , Jun Min Suh , Jiho Shin, Yunpeng Liu, Hanwool Yeon, Kuan Qiao, Hyun S. Kum , Chansoo Kim, Han Eol Lee , Chanyeol Choi , Hyunseok Kim, Doyoon Lee, Jaeyong Lee, Ji-Hoon Kang, Bo-In Park, Sungsu Kang, Jihoon Kim, Sungkyu Kim, J. A. A. J. K. Chip-less wireless electronic skins by remote epitaxial

- freestanding compound semiconductors. *Science (80-.)*. **377**, (2022).
4. Mauthe, S. *et al.* High-speed III-V nanowire photodetector monolithically integrated on Si. *Nat. Commun.* **11**, 1–7 (2020).
 5. Hakkel, K. D. *et al.* Integrated near-infrared spectral sensing. *Nat. Commun.* **13**, 1–8 (2022).
 6. Yang, Z., Albrow-Owen, T., Cai, W. & Hasan, T. Miniaturization of optical spectrometers. *Science (80-.)*. **371**, (2021).
 7. Kuech, T. F. III-V compound semiconductors: Growth and structures. *Prog. Cryst. Growth Charact. Mater.* **62**, 352–370 (2016).
 8. Mandl, B. *et al.* Au-free epitaxial growth of InAs nanowires. *Nano Lett.* **6**, 1817–1821 (2006).
 9. Mandl, B. *et al.* Growth mechanism of self-catalyzed group III-V nanowires. *Nano Lett.* **10**, 4443–4449 (2010).
 10. J. A. Gupta, J. Woicik, S. P. Watkins, D. A. Harrison, E. D. C. and B. A. K. Layer perfection in ultrathin MOVPE-grown InAs layers buried in GaAs(001) studied by X-ray standing waves and photoluminescence spectroscopy. *J. Synchrotron Rad.* **6**, 500–502 (1999).
 11. A. Tabata, T. Benyattou, G. Guillot, M. Gendry, G. H. and P. V. Optical properties of InAs/inP surface layers formed during the arsenic stabilization process. *J. Vac. Sci. Technol. B* **12**, 2299–2304 (1994).